# Automated Glacier Snow Line Altitude Calculation Method Using Landsat Series Images in the Google Earth Engine Platform

Xiang Li [1,2] , Ninglian Wang [1,2,3,*] and Yuwei Wu [1,2]

1 Shaanxi Key Laboratory of Earth Surface System and Environmental Carrying Capacity, Xi'an 710127, China; xli@stumail.nwu.edu.cn (X.L.); htwyw@lzb.ac.cn (Y.W.)
2 Institute of Earth Surface System and Hazards, College of Urban and Environmental Sciences, Northwest University, Xi'an 710127, China
3 Institute of Tibetan Plateau Research, Chinese Academy of Sciences, Beijing 100101, China
* Correspondence: nlwang@nwu.edu.cn; Tel.: +86-29-8830-8526

**Abstract:** Glacier snow line altitude (SLA) at the end of the ablation season is an indicator of the equilibrium line altitude (ELA), which is a key parameter for calculating and assessing glacier mass balance. Here, we present an automated algorithm to classify bare ice and snow cover on glaciers using Landsat series images and calculate the minimum annual glacier snow cover ratio (SCR) and maximum SLA for reference glaciers during the 1985–2020 period in Google Earth Engine. The calculated SCR and SLA values are verified using the observed glacier accumulation area ratio (AAR) and ELA. We select 14 reference glaciers from High Mountain Asia (HMA), the Caucasus, the Alps, and Western Canada, which represent four mountainous regions with extensive glacial development in the northern hemisphere. The SLA accuracy is ~73%, with a mean uncertainty of ±24 m, for 13 of the reference glaciers. Eight of these glaciers yield $R^2 > 0.5$, and the other five glaciers yield $R^2 > 0.3$ for their respective SCR–AAR relationships. Furthermore, 10 of these glaciers yield $R^2 > 0.5$ and the other three glaciers yield $R^2 > 0.3$ for their respective SLA–ELA relationships, which indicate that the calculated SLA from this algorithm provides a good fit to the ELA observations. However, Careser Glacier yields a poor fit between the SLA calculations and ELA observations owing to tremendous surface area changes during the analyzed time series; this indicates that glacier surface shape changes due to intense ablation will lead to a misclassification of the glacier surface, resulting in deviations between the SLA and ELA. Furthermore, cloud cover, shadows, and the Otsu method limitation will further affect the SLA calculation. The post-2000 SLA values are better than those obtained before 2000 because merging the Landsat series images reduces the temporal resolution, which allows the date of the calculated SLA to be closer to the date of the observed ELA. From a regional perspective, the glaciers in the Caucasus, HMA and the Alps yield better results than those in Western Canada. This algorithm can be applied to large regions, such as HMA, to obtain snow line estimates where manual approaches are exhaustive and/or unfeasible. Furthermore, new optical data, such as that from Sentinel-2, can be incorporated to further improve the algorithm results.

**Keywords:** snow line altitude (SLA); Landsat; glacier; equilibrium line altitude (ELA); Google Earth Engine (GEE)

## 1. Introduction

Most of the world's glaciers are currently experiencing mass loss, which will contribute to a host of global environmental problems, such as increasing glacier meltwater, rising sea level, and increasing water scarcity [1–3]. Mountain glaciers form a critical component of the cryosphere and are sensitive to climate change. The glacier equilibrium line altitude (ELA), which is representative of the annual glacier mass balance and the response of that glacier to climate change, is defined as the glacier position where its annual ablation equals

accumulation. The ELA is generally determined via field mass-balance measurements, such as snow pits and stakes [4]. However, this field-based approach is unfeasible for monitoring many mountain glaciers due to their remote environments and limited observation sites [5]. Cuffey and Paterson [6] have highlighted that the snow line altitude (SLA) at the end of the ablation season is a good approximation of the ELA. The SLA is defined as the glacier position that marks the maximum surface extent of snow cover, with bare ice present below the SLA and snow cover present above the SLA. Therefore, observations of the glacier SLA can be used to assume the ELA [4,7–9].

Remote sensing is an effective approach for mapping glacier SLA using high-resolution spatiotemporal optical images. Several datasets, such as remote-sensing images, glacier boundary shapefiles, and digital elevation models (DEMs), are employed to obtain the glacier SLA. Landsat series images (Landsat Thematic Mapper (TM), Enhanced Thematic Mapper Plus (ETM+), Operational Land Imager (OLI)) are the ideal optical images for calculating the SLA in the long period. Although Landsat images possess a lower temporal resolution than Moderate Resolution Imaging Spectroradiometer (MODIS) images and are more susceptible to cloud cover and shadow than synthetic aperture radar (SAR) images, they are acquired at a high spatial resolution and span a long temporal range (1984 to present). Reference glacier inventories that have been published by the World Glacier Monitoring Service (WGMS, https://wgms.ch/, accessed on 9 February 2022) and National Cryosphere Desert Data Center (NCDC, http://www.ncdc.ac.cn/, accessed on 2 May 2022) provide key glacier parameters, such as the glacier outline, area, volume, length, and ELA for SLA analyses.

Accurate bare ice and snow cover on glacier classifications are critical constraints during the SLA calculation process and several methods have been proposed for comprehensive and accurate glacier classification. The normalized difference snow index (NDSI), which is calculated as the normalized difference between the green and shortwave infrared (SWIR) bands, has been widely used for snow mapping [10–12]. Band ratio methods, such as the near infrared band (NIR)/SWIR or red band/SWIR methods, offer another approach for identifying snow cover on glaciers [13–15]. However, these methods are more adept for classifying snow cover together rather than classifying bare ice and snow cover on glaciers individually. The widely used approach for mapping snow cover and bare ice on a glacier is to seek a threshold where the snow is obviously different from the ice. The NIR band is a proven wavelength for mapping snow cover and bare ice because of their albedo difference in the NIR band [16–18]. The use of a narrow band that is converted to a broad band and then employed to seek a threshold for bare ice and snow cover classification is another method [19–22].

The glacier SLA can then be calculated after the glacier surface classification. We note that the SLA is rarely a continuous and apparent line that separates bare ice and snow cover, as random snow patches are often present near the SLA, thereby blurring the true snow line. Therefore, a robust and automated algorithm is crucial for adapting to this challenge of determining glacier SLA. Three types of methods are generally used to delineate the SLA, i.e., the point, line, and zone methods. For the point method, Krajčí et al. [23] proposed that SLA is defined as the elevation where there are minimum snow pixels below it and maximum ice pixels above it. For the line method, the SLA is considered and represented as a line on the graph, with the snow area above the line and bare ice area below the line. Manual SLA delineation is the most widely used line method by visual interpretation on optical image [11]. It often has higher accuracy than an automated calculation on glaciers with distinct boundaries between ice and snow cover. However, the zone method may be more robust and suitable for automated calculations due to its tolerance of potential snow patches near the SLA. The accumulation area ratio (AAR) method, which was first proposed by Meier [4], can assume the glacier SLA based on an input AAR value. Specifically, if the selected area is larger than the snow cover area (glacier area × AAR), then the altitude where this area located is considered the SLA. The Accumulation Area Ratio (AAR) is the ratio of the area of the accumulation zone to the entire glacier. The AAR and ELA

have strong correlations on many glaciers and glaciers in balance have stable AAR values ranging from 0.5 to 0.8, generally [4]. The AAR method has become a widely used method for estimating ELA [24–26]. The World Meteorological Organization (WMO) has proposed identifying the zone where there is >50% snow cover, as this zone's average altitude represents the SLA. Rastner et al. [27] used this method and identified >50% snow cover in five consecutive zones (20-m intervals), with the altitude of the lowest zone representing the SLA.

All of the satellite images during the ablation period should be analyzed to obtain the annual glacier SLA. However, Landsat has a 16-day temporal resolution, and the combination of images from two different Landsat missions (such as Landsat-7 and Landsat-8) can only improve the temporal resolution to 8 days. Furthermore, some images cannot be selected for analysis due to cloud cover, cloud shadow, and/or fresh snow effects. Previous studies [12,22,28–30] have manually selected suitable images for glacier SLA calculations; however, although these SLA calculations may provide an accurate SLA for the selected image day, they may not capture the day of highest SLA, which is required to accurately represent the ELA.

Here we present an automated SLA algorithm and apply it to Landsat images to (a) map the bare ice and snow cover on a given glacier, and (b) calculate and construct the glacier SLA time series. We use 14 reference glaciers to test and validate the algorithm. The entire process is executed in Google Earth Engine (GEE), a remote-sensing big data platform. Our method (a) extracts all of the images during the ablation period and selects the available images for further calculation; (b) uses an enhanced band, which is a combination of the NIR band and the NIR/SWIR ratio, to create a threshold for classification via the Otsu [31] method; and then (c) calculates the glacier SLA based on the WMO method to generate the glacier SLA time series. The resultant SLA data are validated using field ELA data.

## 2. Study Area and Data

### 2.1. Study Area

This study focuses on four regions where mountain glaciers are common: High Mountain Asia (HMA), the Caucasus, the Alps, and Western Canada (Figure 1). Fourteen glaciers are selected as reference glaciers, with each possessing more than 30 years of ongoing ELA measurements that can be used for validation.

Four glaciers are selected for SLA calculations in HMA: Qiyi, Urumqi No. 1, Abramov, and Tsentralniy Tuyuksuyskiy glaciers. HMA is located in central Asia and extends from five central Asian countries in the west to China in the east, with approximately 100,000 identified mountain glaciers in this region [32–34].

Qiyi Glacier (39.24°N, 97.76°E), which is located in the Qilian Mountains, China, is the first mountain glacier to be studied by Chinese researchers in 1958. Qiyi Glacier covers an area of 2.76 km$^2$ and extends from 4304 m above sea level (a.s.l.) to 5159 m a.s.l. Measurements were acquired at Qiyi Glacier from 1958 to 2016.

Urumqi Glacier No. 1 (43.08°N, 86.82°E), which is located in Tien Shan, China, has been studied for nearly 65 years. It consists of two separate branches, the east and west branches of Glacier No. 1. The east branch covers an area of 1.07 km$^2$ and extends from 3743 to 4484 m a.s.l. The west branch covers an area of 0.58 km$^2$ and extends from 3743 to 4267 m a.s.l. Measurements were acquired at Urumqi Glacier No. 1 from 1959 to present.

Abramov Glacier (39.61°N, 71.56°E), which is a valley glacier in Pamir-Alay, Kyrgyzstan, covers an area of ~24 km$^2$ and extends from 3650 to nearly 5000 m a.s.l. The WGMS glacier inventory indicates that measurements were acquired at Abramov glacier from 2010 to 2019.

Tsentralniy Tuyuksuyskiy Glacier (43.05°N, 77.08°E), which is also called Tuyuksu Glacier, is located in the Zailiyskiy Alatau Range of the Kazakh Tien Shan, Kyrgyzstan. It covers an area of 2.3 km$^2$ (measured in 2013) and extends from 3478 to 4219 m a.s.l. Measurements were acquired at Tsentralniy Tuyuksuyskiy Glacier from 1958 to 2019.

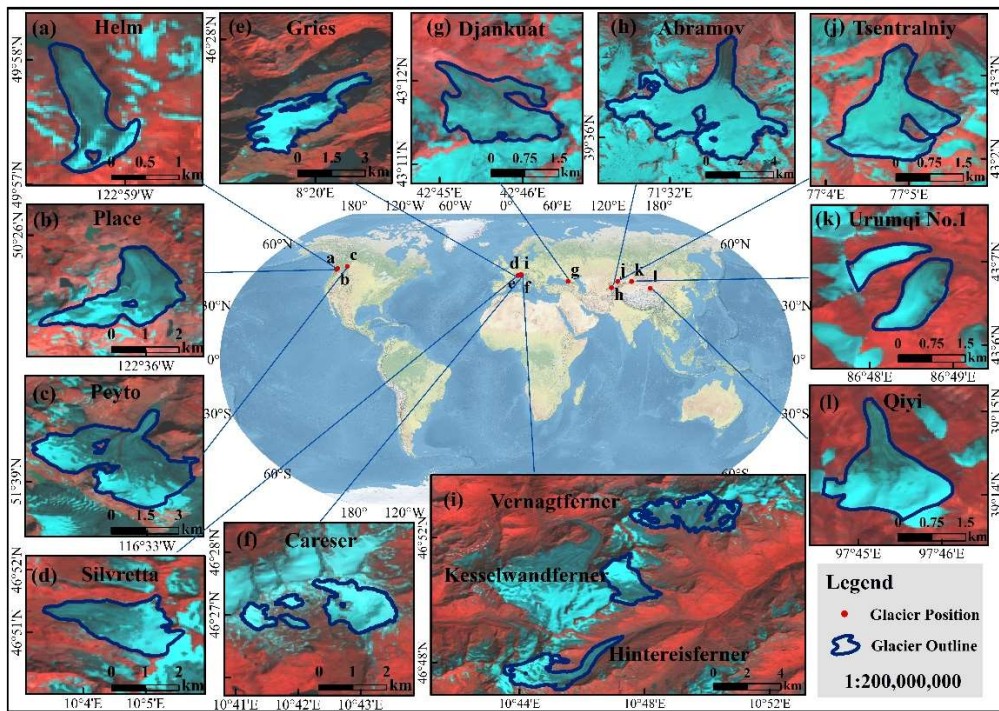

**Figure 1.** Study areas and their global distribution. Glacier images are Landsat series images (false-color composite in the SWIR, blue, and green bands).

A single reference glacier is selected in the Caucasus. Djankuat Glacier (43.20°N, 42.77°E) is a northwest-facing, debris-covered valley glacier that is located near the border of the Russian Federation with Georgia. It covers an area of 2.69 km$^2$ and extends from 2700 to 3750 m a.s.l. Measurements were acquired at Djankuat Glacier from 1968 to 2009.

Six reference glaciers, Silvretta, Gries, Careser, Hintereisferner, Kesselwandferner, and Vernagtferner glaciers, are selected across the Alps for the SLA calculations. The glaciers across the European Alps have experienced accelerated glacier shrinkage and negative mass balances since the 1980s, which have also been observed across glaciers worldwide [30–32].

Silvretta Glacier (46.85°N, 10.08°E) is a mountain glacier that is located in northeastern Switzerland, near the border with Austria. It covers an area of ~2.70 km$^2$ and extends from 2470 to 3070 m a.s.l. Measurements were acquired at Silvretta Glacier from 1919 to 2020.

Gries Glacier (46.44°N, 8.34°E) is a temperate valley glacier located in the central Swiss Alps at the border to Italy. It covers an area of ~4.40 km$^2$ and extends from 2430 to 3010 m a.s.l. Measurements were acquired from 1961 to present.

Careser Glacier (46.45°N, 10.70°E) is located in Eastern European Alps, Italy. It covers an area of ~4.70 km$^2$ (in 1967) and reduces to ~1.90 km$^2$ (in 2010) due to strongly ablation. It extends from 2880 to 3278 m a.s.l. Measurements were acquired from 1967 to 2020.

Hintereisferner (46.80°N, 10.77°E), Kesselwandferner (46.84°N, 10.79°E), and Vernagtferner (46.88°N, 10.82°E) glaciers are three mountain glaciers that are located in the Eastern Alps of Austria. Hintereisferner covers an area of 7.80 km$^2$ (measured in 2003), extends from 2507 to 3739 m a.s.l., and has continuous long-term observations from 1953 to 2020. Vernagtferner covers an area of ~8.80 km$^2$, extends from 2770 to 3630 m a.s.l., and has continuous observations from 1965 to 2019. Kesselwandferner covers an area of 3.90 km$^2$, extends from 2700 to 3500 m a.s.l., and has long-term observations from 1953 to 2020 (2015–2017 data are missing).

The last three reference glaciers, Helm, Place, and Peyto, are from Western Canada. Observations have been acquired across more than 50 glaciers from western North America, but continuous records have only been acquired at some of these glaciers in the 21st century.

Peyto Glacier (51.67°N, 116.53°W) is located in the southern Canadian Rocky Mountains. It covers an area of 11.40 km$^2$, extends from 2100 to 2640 m a.s.l., and has nearly 50 years of observations.

Place (50.40°N, 122.60°W) and Helm (49.97°N, 123.00°W) glaciers are located in the Canadian Coast Mountains. Place covers an area of ~3.98 km$^2$, extends from 1800 to 2610 m a.s.l., and has observations from 1965 to 2019. Helm covers an area of ~0.80 km$^2$, extends from 1700 to 1900 m a.s.l., and has observations from 1975 to 2019.

In addition to Qiyi Glacier, other glaciers have published ELA and AAR datasets. Qiyi Glacier has ELA measurement data. Each glacier's geographical properties are shown in Table 1.

**Table 1.** Geographical properties of reference glaciers. $Z_{min}$ and $Z_{max}$ represent the minimum and maximum height of the glacier. $L_{max}$ refers to the longest length of the glaciers.

| Name | Central Lon | Central Lat | Region | Area/km$^2$ | $Z_{min}$/m a.s.l. | $Z_{max}$/m a.s.l. | Aspect/° | Slope/° | $L_{max}$/m | Published Observe Period |
|---|---|---|---|---|---|---|---|---|---|---|
| Helm | 123.00°W | 49.97°N | Western Canada | 0.80 | 1700 | 1900 | 306 | 12.4 | 2143 | 1975–2019 |
| Place | 122.60°W | 50.40°N | Western Canada | 3.98 | 1800 | 2610 | 9 | 11.1 | 3589 | 1965–2019 |
| Peyto | 116.53°W | 51.67°N | Western Canada | 11.40 | 2100 | 2640 | 29 | 12.7 | 5387 | 1966–2019 |
| Silvretta | 10.08°E | 46.85°N | Alps | 2.70 | 2470 | 3070 | 292 | 12.9 | 3151 | 1919–2020 |
| Gries | 8.34°E | 46.44°N | Alps | 4.40 | 2430 | 3010 | 41 | 11.8 | 5652 | 1961–2020 |
| Careser | 10.70°E | 46.45°N | Alps | 1.90 | 2880 | 3278 | 176 | 10.6 | 2305 | 1967–2020 |
| Djankuat | 42.77°E | 43.20°N | Caucasus | 2.69 | 2700 | 3750 | 322 | 21.9 | 3043 | 1968–2009 |
| Abramov | 71.56°E | 39.61°N | HMA | 24 | 3650 | 5000 | 10 | 27.0 | 8572 | 2010–2019 |
| VNF * | 10.82°E | 46.88°N | Alps | 8.80 | 2770 | 3630 | 165 | 14.7 | 3087 | 1965–2019 |
| KWF * | 10.79°E | 46.84°N | Alps | 3.90 | 2700 | 3500 | 123 | 11.6 | 4202 | 1953–2020 |
| HEF * | 10.77°E | 46.80°N | Alps | 7.80 | 2507 | 3739 | 71 | 16.2 | 7178 | 1953–2020 |
| Tsentralniy | 77.08°E | 43.05°N | HMA | 2.30 | 3478 | 4219 | 359 | 19.1 | 3131 | 1958–2019 |
| Urumqi No. 1 | 86.82°E | 43.08°N | HMA | 1.65 | 3743 | 4484 | 33 | 20.6 | 1946 | 1959–2020 |
| Qiyi | 97.76°E | 39.24°N | HMA | 2.76 | 4304 | 5159 | 2 | 19.2 | 3007 | 1958–2016 |

* VNF, KWF, HEF are Vernagtferner, Kesselwandferner and Hintereisferner, respectively.

### 2.2. Analyzed Data

#### 2.2.1. Landsat Series Data

Nine Landsat missions have been successfully launched since the first Landsat satellite launch in 1972 and millions of images have been obtained. Here, we select Landsat surface reflectance (SR) images with atmospheric and orthographic corrections. A total of 2684 Landsat images were selected to calculate SLA in GEE for the 1985–2020 period (1 July–31 September timeframe) and these selected images are subsets of the total images in time period for the SLA calculations after filtering by cloud cover, shadows and other factors. Here, a given image is retained for analysis if >65% of the glacier surface is visible after masking the cloud cover and shadow effects. Furthermore, a given year is considered a valid year for SLA calculation when more than three images are selected for that year. Therefore, the SLA time series may not be continuous. The number of selected images for each glacier is listed in Table 2.

**Table 2.** Number of selected images for each reference glacier.

| Glacier Name | Selected Images | Glacier Name | Selected Images | Glacier Name | Selected Images |
|---|---|---|---|---|---|
| Helm | 273 | Careser | 110 | Hintereisferner | 213 |
| Place | 231 | Djankuat | 241 | Tsentralniy | 97 |
| Peyto | 197 | Abramov | 252 | Urumqi No. 1 | 116 |
| Silvretta | 331 | Vernagtferner | 136 | Qiyi | 125 |
| Gries | 213 | Kesselwandferner | 149 | Total | 2684 |

### 2.2.2. NASADEM

NASADEM is generated from the Shuttle Rader Topography Mission (SRTM) data, relying on the latest unwrapping techniques and auxiliary data. Advanced Spaceborne Thermal Emission and Reflection Radiometer (ASTER), Advanced Long Observing Satellite (ALOS) and Panchromatic Remote-sensing Instrument for Stereo Mapping (PRISM) datasets are used to fill the voids in NASADEM. The vertical and tilt adjustments are based on ground control points and the Ice, Cloud, and Land Elevation Satellite (ICE-Sat). The accuracy of NASADEM is not well-reported in the literature since this new dataset has only been recently released to the public (in 2020). Bettiol et al. [35] presented the horizontal and vertical accuracies of NASADEM by their calculation. Here, the NASADEM parameters are derived from United States Geological Survey (USGS, https://lpdaac.usgs.gov/products/nasadem_hgtv001/, accessed on 2 May 2022) and accuracy is derived from Bettiol's research (Table 3).

**Table 3.** NASADEM parameters.

| Dataset | Temporal Extent | Spatial Extent | Pixel Size | Horizontal Accuracy | Vertical Accuracy |
|---|---|---|---|---|---|
| NASADEM | 2000-02-11 to 2000-02-21 | 60°N to 56°S, 180°W to 180°E | 30 m | 20 m | 16 m |

### 2.2.3. Glacier Data

Glacier shapefiles were downloaded from the WGMS and NCDC websites, and they provided the glacier outlines, AARs, and ELAs. However, some of the glacier outlines required modification owing to glacier variations and/or unmatched geometries. We used the Randolph Glacier Inventory 6.0 (RGI 6.0) as a reference, a global inventory of glacier outlines that also containing all of the reference data, to revise some of the WGMS and NCDC data and confirm the glacier outline accuracy. We ultimately selected glacier outlines from the 2010–2016 period and manually revised some of them to generate the final glacier feature collection owing to temporal variations in some of the glacier outlines.

## 3. Methods

The entire flowchart for the image processing and SLA calculation is shown in Figure 2. There are three main input datasets: the Landsat SR images, glacier outline shapefiles, and NASADEM. The processing can be divided into two loops. Each year is a variable that controls the loop period in the outer loop, and all of the images are filtered and selected for calculation in the inner loop. The algorithm can then determine the minimum SCR and maximum SLA. The annual glacier SCR and SLA can be calculated and validated using the AAR and ELA, respectively, with these two loops. This automated algorithm is written in JavaScript in GEE, and all of the processing is automated, with only three inputs required. The specific steps are presented as follows.

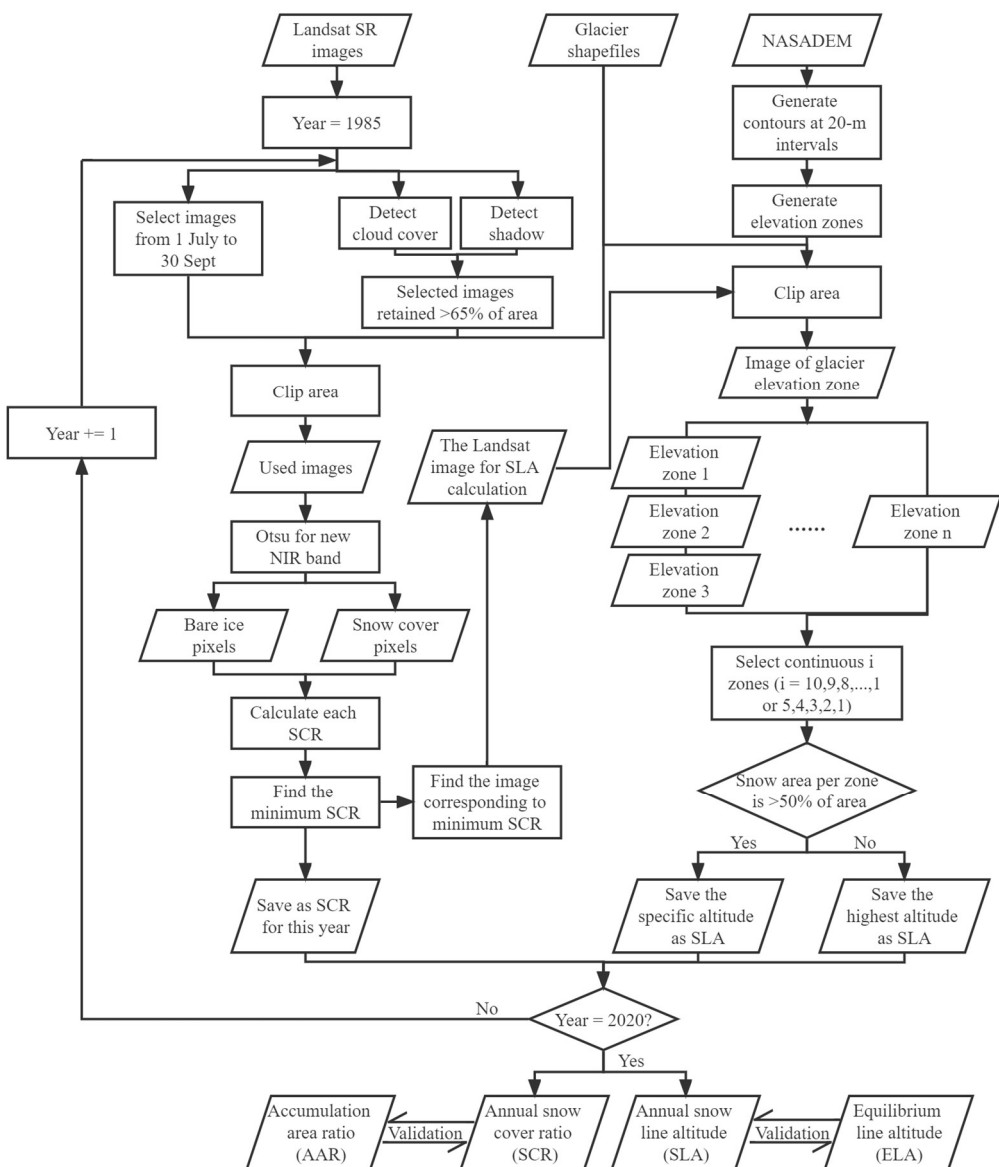

**Figure 2.** Processing flowchart for the image processing and SLA calculation.

### 3.1. Preprocessing

#### 3.1.1. Rename Landsat Bands

Different sensors have been carried onboard the Landsat series of satellites (Landsat TM, ETM+, OLI), which has led to acquired images at the same wavelength for a given band that possess different band designations. For example, the blue band in Landsat 5 and Landsat 7 is B1, whereas it is B2 in Landsat 8. It is therefore necessary to unify these bands and rename them to avoid potential processing errors. Here we select seven bands for analysis: the blue, green, red, NIR, SWIR 1 and 2 and QA bands. The analyzed series of Landsat images can be regarded as homogeneous images for the SLA calculations by selecting and renaming the bands. This selection and renaming process is shown in Table 4.

**Table 4.** Band renaming for the analyzed Landsat series images.

| Landsat-5 | Landsat-7 | Landsat-8 | Image Collection |
|-----------|-----------|-----------|------------------|
| B1 | B1 | B2 | Blue |
| B2 | B2 | B3 | Green |
| B3 | B3 | B4 | Red |
| B4 | B4 | B5 | Nir |
| B5 | B5 | B6 | Swir1 |
| B7 | B8 | B7 | Swir2 |
| QA_PIXEL | QA_PIXEL | QA_PIXEL | QA |

### 3.1.2. Cloud Cover and Shadow Detection

Cloud cover, shadow, and terrain are significant factors that lead to inaccurate classification results. Therefore, the accurate detection of pixels containing cloud cover and shadow is critical for determining the image quality. The quality assessment (QA) band, which includes cloud cover, cloud shadow, and other parameters, is calculated by the USGS via the CFMask algorithm in GEE. CFMask is a multi-pass algorithm that is derived from the Function of Mask (FMask), which was written at Boston University and translated to C at USGS EROS [36–39]. The cloud cover, snow cover, cloud shadow, and other ground features are assessed and stored as bit values in GEE, which can then be used to detect the cloud cover and cloud shadow pixels. However, CFMask may have difficulties over bright targets, such as snow and ice, which means that cloud cover and cloud shadow detection via CFMask is not definitive. Here, we define a suitable image for SLA calculation as one that retains >65% of its glacier area after detecting and removing any cloud cover and shadow pixels.

### 3.2. Generate Used Images

The original Landsat SR images cannot be used to directly calculate the SLA. All of the potential images are merged in an ImageCollection (a type in GEE) after data unification and band renaming. The suitable images can then be filtered by ImageCollection after detecting the cloud cover and cloud shadow via CFMask. The image extent is clipped by the glacier outline in the following step, such that only the area covering the glacier is retained. The ImageCollection dataset, which contains the images to be used for the glacier cover classification, is then generated.

### 3.3. Glacier Surface Classification and Snow Cover Ratio (SCR) Calculation

The Otsu method is employed to classify the used Landsat SR images and generate snow cover and bare ice pixels in GEE. Previous studies have demonstrated that the joint identification of snow and ice on a glacier can be easily distinguished via either the NDSI or band ratio methods, such as NIR/SWIR [13,14]. However, it is difficult to separate bare ice and snow cover on glacier via these methods due to the similar reflectances in these bands. Previous studies have generated histograms of the NIR band pixels to distinguish snow cover and bare ice due to their different brightness temperatures [27]. Here we find that combining the NIR band and band ratio methods can enhance the classification results (Equation (1)). Therefore, we create a new single band to generate a histogram for the glacier cover classification.

$$\text{NIR}_{\text{NEW}} = \text{NIR} \times (\text{NIR}/\text{SWIR}) \tag{1}$$

Figure 3 shows two relatively extreme situations for the snow and ice classification: fresh snow covering the entire glacier and snowmelt across most of the glacier. In the first situation, we find that the NIR band classification yields more misclassifications than new NIR band classification. The true SCR is close to 1. The SCR for classification using $\text{NIR}_{\text{NEW}}$ is 0.90, while the SCR for classification using NIR is 0.68, so the accuracy is higher when using $\text{NIR}_{\text{NEW}}$ for classification. In the latter situation, the two methods yield similar

results, and both of their SCR is 0.28. Overall, these results demonstrate that the new NIR band can enhance accuracy of the classification results.

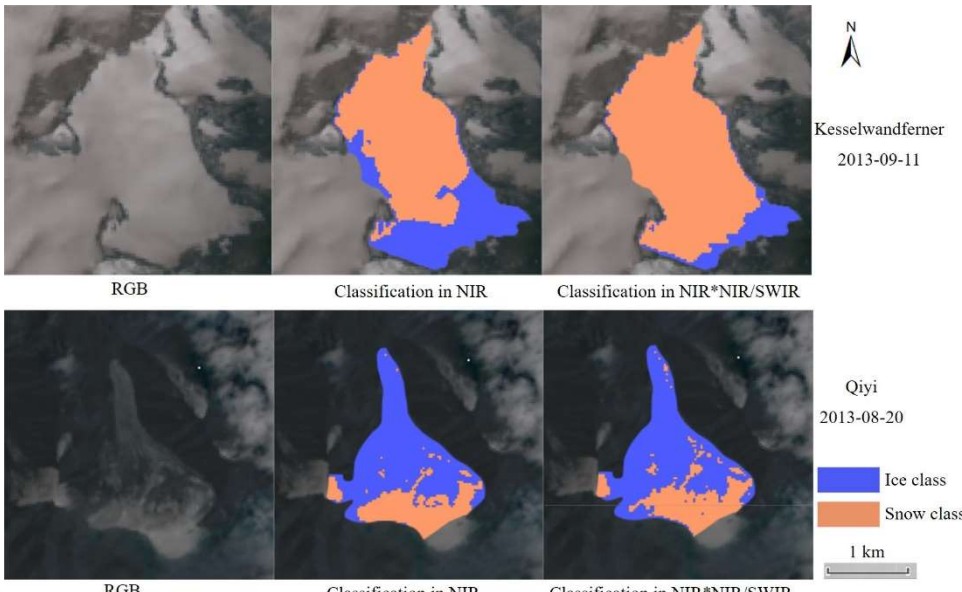

**Figure 3.** Glacier surface classification results using the NIR band and new NIR band (NIR × NIR/SWIR).

Mapping the snow and ice classification is the first step in determining the SCR. We follow the approach presented by Donchyts et al. [40] to code the Otsu method in GEE. We assume that each used image only contains two classes which are bare ice and snow cover on glacier. The Otsu method calculates the intensity threshold for each image pixel to find the maximum inter-class variance. Several types of cover are generally observed on glaciers (snow, ice, debris, rock, and melt water), which means more than two types should be classified. However, snow and ice dominate the glacier area, with the other features being grouped into these two types. Debris and rock are dark, so they are classified as bare ice, and meltwater is classified as snow cover based on its bright condition. However, extreme scenarios may arise, where snowfall occurs a few days before image acquisition, such that fresh snow covers the entire glacier. Only one surface type exists on the glacier, but the Otsu method still generates two types by introducing an error that classifies some dark extents as bare ice. However, this error will not affect the final SCR calculation because this SCR is larger than the other SCR values, and it will not be selected as the minimum SCR in the analyzed year. Another extreme scenario is the removal of almost all of the snow cover via snowmelt, such that bare ice extends along the entire glacier. Some bright areas will be classified as snow cover, which will yield an SLA. This SLA may represent the maximum SLA in this year, but it will be lower than the ELA because misclassification of snow cover reduces the real SLA.

The SCR for each image is calculated after the snow and ice classification (Equation (2)). Every result is analyzed to find the minimum SCR in a given year (Equation (3)), with the bare ice and snow cover extents that produced this minimum SCR used for the SLA calculation.

$$SCR = Snow\ Pixels/(Snow\ Pixels + Ice\ Pixels) \qquad (2)$$

$$SCR_{thisYear} = SCR_{minimum} \qquad (3)$$

### 3.4. Snow Line Altitude (SLA) Calculation

The image and classification results that are associated with the minimum SCR values are used for the SLA calculations for each glacier. The snow line across a given glacier is ideally a continuous line that forms the boundary between snow and ice. The glacier snow line is the instantaneous location of the snow line on an optical image of the glacier.

However, blowing snow, fresh snow and shadows may blur the exact position of the snow line, such that a mixed zone of snow cover and ice is identified instead of an obvious line. Several methods have been derived for snow line calculations, with these approaches generally classified as point, line, and zone methods. Here we use a zone method that was created by the WMO. The key to this method is to seek zones where there is >50% snow cover. A DEM is first employed to generate 20-m contours. This contour map is then used to clip the classification images and generate zones for each 20-m interval. WMO suggests that the number of continuous zones should be selected depending on the glacier size and the problem to be treated. Rastner et al. [27] used five continuous zones (100 m, 20 m per zone) to calculate the SLA. Here we propose the number based on experiments that five continuous zones are used if the glacier area is <10 km$^2$ (100 m) and eight if the area is >10 km$^2$ (160 m). The algorithm first selects every set of five (or eight) continuous zones list for calculation; if each zone possesses >50% snow cover, then the lowest altitude is regarded as the SLA. This algorithm traverses the entire glacier from low to high until it finds a match condition. If there are not five continuous zones that satisfy the above condition, then the number of zones is reduced to four (or seven); this number is reduced until the above condition is met. The algorithm regards the highest altitude of the glacier as the SLA if all of the conditions are not satisfied in the calculation.

The SLA calculations for Qiyi (<10 km$^2$ area) and Abramov (>10 km$^2$ area) glaciers are shown in Figure 4. The two red arrows on each image indicate the continuous five (Qiyi) and eight (Abramov) zones. Snow patches, rock, edge areas, and a few misclassifications from this method do not affect the final SLA calculation result, which highlights the good tolerance of this method.

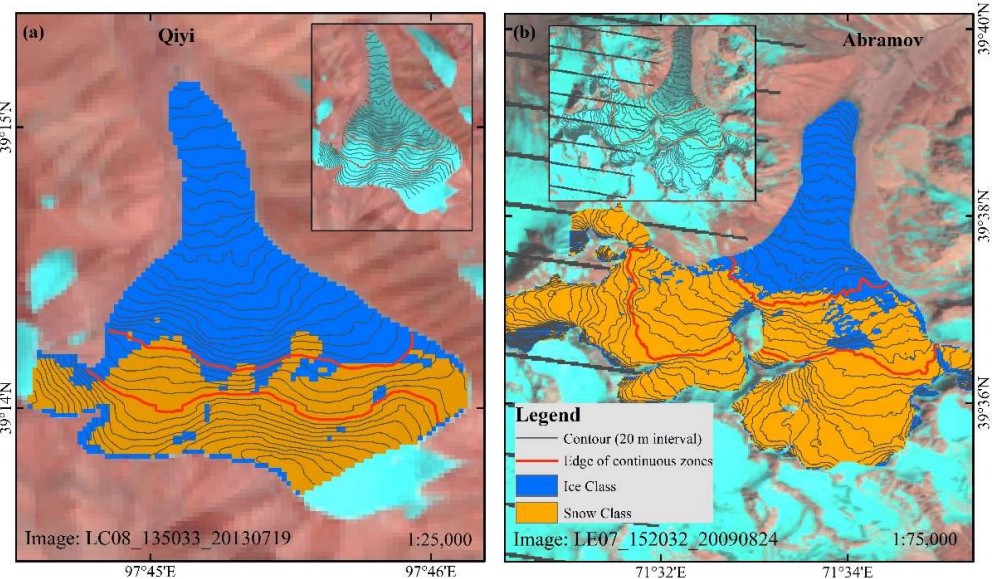

**Figure 4.** SLA calculations for two reference glaciers. (**a**) Qiyi Glacier. The area is <10 km$^2$, with the two red lines highlighting the highest and lowest altitudes of five continuous zones where there is >50% snow cover. (**b**) Abramov Glacier. The area is >10 km$^2$, with the two red lines highlighting the highest and lowest altitudes of eight continuous zones where there is >50% snow cover. Some edge misclassifications exist, but these do not affect the SLA calculations. The background images are Landsat images with false color combination in the SWIR, green and blue bands.

When each SLA value is derived, it represents the SLA at image acquisition day. In the ablation season, all calculated SLA values build up the SLA sequences, and the maximum SLA is selected as SLA this year. For instance, Figure 5 shows Qiyi Glacier SLA sequences in 2015. With filtering and selecting by the above rules, 6 of 12 images in the time period are generated as used images for calculating SLA. After each SLA value calculation, the maximum SLA is derived as 4778 m a.s.l., which is acquired on 18 August.

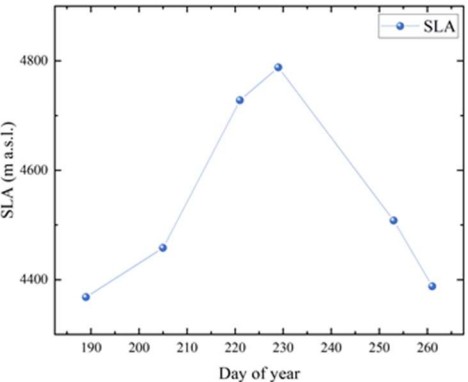

**Figure 5.** Qiyi Glacier SLA sequences in 2015. Six blue dots represent six SLA values during the ablation season.

### 3.5. Uncertainty Estimates

The uncertainty in the SLA calculation arises from the vertical accuracy of the DEM, the accuracy of the glacier outline, and the method uncertainty. The algorithm is fully automated in this study, such that any potential human errors are minimized. Paul et al. [41] determined that the accuracy of the remote-sensing-derived glacier data is based on the pixel resolution, with an estimated uncertainty of 1 pixel size. The vertical accuracy of the DEM is derived from Bettiol et al. [35]. The method uncertainty arises during the snow and ice classification and the zone delineation for the SLA calculation. However, classifications with mixed pixels are inevitable and inestimable. Therefore, the uncertainties are defined and estimated as follows: (a) $u_{DEM}$ is the vertical error of NASADEM. Here, we define the uncertainty as $\pm 16$ m. (b) $u_{outline}$ indicates the accuracy of the glacier outlines, which is estimated to be 1/2 pixel. The analyzed Landsat series images possess a 30-m resolution in the NIR and SWIR bands. Therefore, the associated uncertainty is $\pm 15$ m. (c) $u_{method}$ arises during the glacier SLA calculation, as the glacier is divided into 20-m zones; we set the mean elevation of each zone as the potential SLA. Therefore, the associated uncertainty is $\pm 10$ m. The total uncertainty ($u$) is estimated via the root mean square error formula, yielding:

$$u = \pm\sqrt{u_{DEM}^2 + u_{outline}^2 + u_{method}^2} = \pm 24 \text{ m} \tag{4}$$

### 4. Results

#### 4.1. Glacier Snow Cover Ratio (SCR) Time Series

The annual minimum SCR and annual AAR for 13 of the reference glaciers (the Qiyi AAR data are not published) are presented in Figure 6, along with their respective linear time-series trends. The AAR time series are generally lower than the SCR time series, which means that the true ablation is often stronger than the obtained SCR values from the image calculations due to the Landsat temporal resolution and weather effects. Helm, Place, Peyto, Silvretta, Gries, Djankuat, Vernagtferner, Kesselwandferner, Hintereisferner, and Urumqi No. 1 glaciers possess similar AAR and SCR linear trends, which means that the long-term SCR time series can represent the glacier mass balance, and can therefore be used fill data gaps when AAR data are missing. However, no such similarity between the AAR and SCR linear trends is observed for Careser, Abramov, and Tsentralniy glaciers. Careser Glacier has experienced intense ablation over the past 40 years, such that its AAR value is constant at approximately 0 and its surface shape has undergone significant changes. The algorithm calculates SCR values that are similar to the measured AAR values for Careser Glacier in 2000, 2010, 2014 and 2015, but the SCR and AAR values are quite dissimilar for the other years. The sparse AAR and SCR data for Abramov Glacier likely contribute to the different AAR and SCR linear trends, as similar trends are observed for every year except 2012 and 2019. The SCR linear trend for Tsentralniy Glacier is affected by the 2002 and 2003 values,

which are quite different from the corresponding AAR values; the SCR trend is consistent with AAR trend when the 2002 and 2003 SCR values are removed. We note that the AAR values can reach 0 on some glaciers owing to intense ablation, whereas the obtained SCR values are larger than 0 due to the Otsu method and Landsat image limitations.

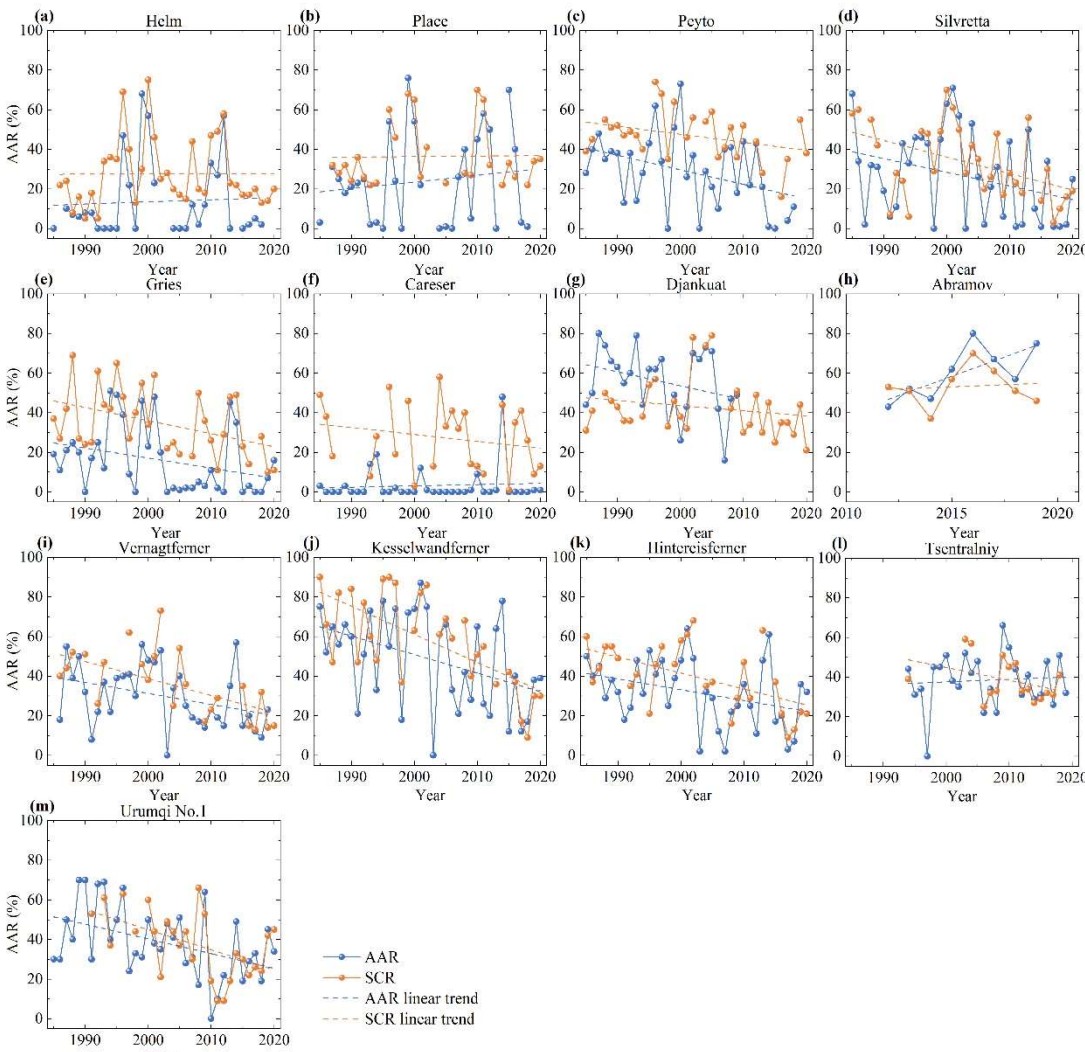

**Figure 6.** AAR and SCR time-series changes and linear trends. Blue and orange dots represent the AAR and SCR values, respectively. Solid lines highlight the time-series changes, and dashed lines represent the linear trends. Subfigures (**a**–**m**) represent each value of SCR and AAR per year, and the time trend between the SCR and AAR of each glacier, respectively.

A comparison of the minimum SCR values and the AAR values for 13 of the reference glaciers is shown in Figure 7. Silvretta Glacier possesses the highest $R^2$ value at 0.62 and significantly correlates at the 0.001 level, which means that the obtained SCR provides an excellent fit for the AAR. Helm, Place, Vernagtferner, Kesselwandferner and Hintereisferner glaciers possess $R^2$ values above 0.5 and significantly correlate at the 0.001 level. Peyto, Gries and Urumqi No. 1 glaciers possess $R^2$ values above 0.4 and significantly correlate at the 0.001 level. Besides, Djankuat and Tsentralniy glaciers possess an $R^2$ value above 0.5 and significantly correlate at the 0.01 level. These glaciers yield a good-fitting SCR-AAR relationship. However, Careser and Abramov glaciers have not passed the significant test. Careser Glacier possesses an $R^2$ value of only 0.01, which highlights the large deviations between the obtained SCR and AAR values. Abramov possesses an $R^2$ value of 0.34 and

a relatively lower R² and higher p than other glaciers due to sparse SCR-AAR pairs. The differences in 2012 and 2019 affect the relationship between SCR and AAR on Abramov.

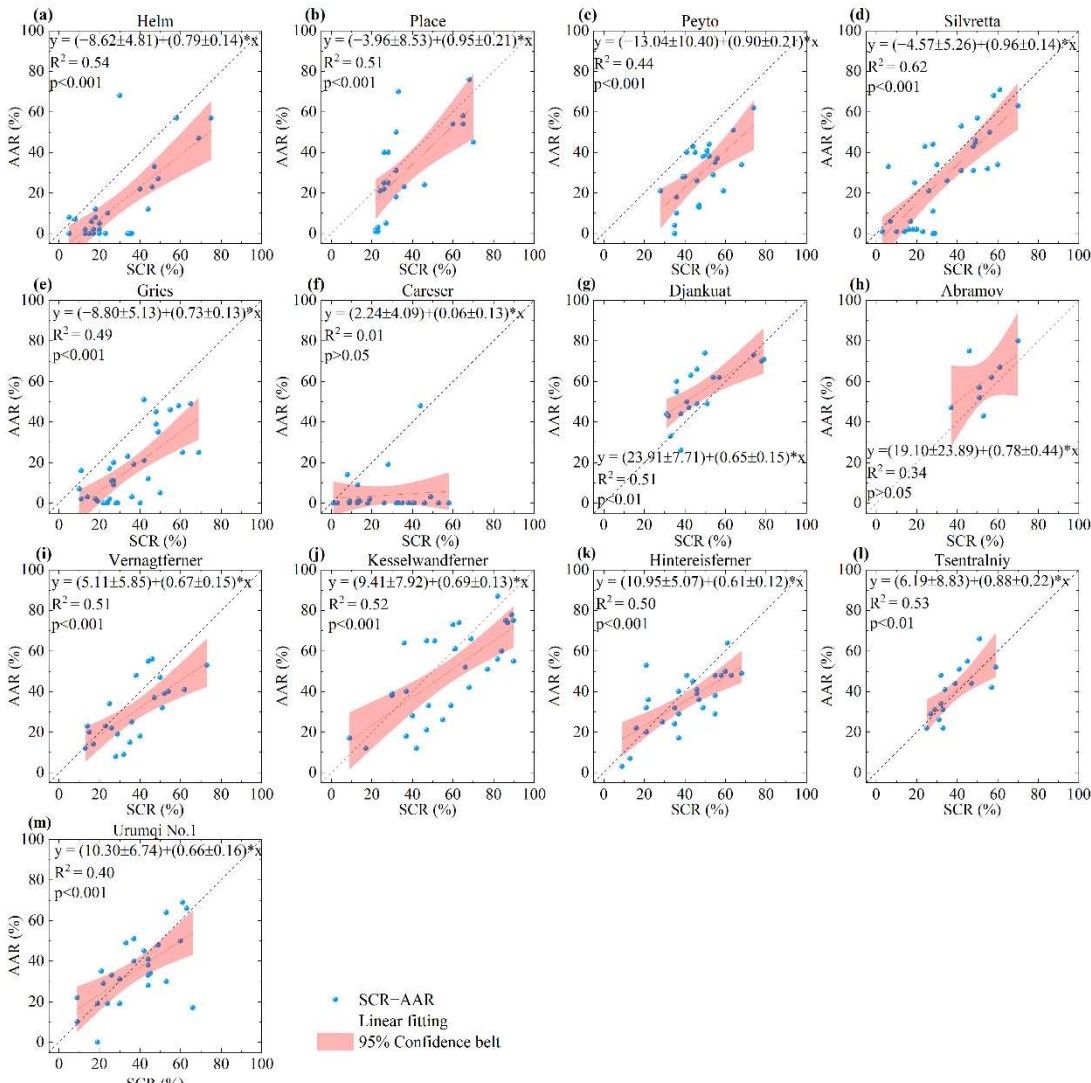

**Figure 7.** SCR–AAR relationship for 13 of the reference glaciers. Blue points represent the annual SCR–AAR pairs, solid lines indicate the linear fits to the data, and red areas are the 95% confidence belts. Subfigures (**a**–**m**) represent the comparison of the relationship between the SCR and AAR of each glacier, respectively. The linear function equation in each subfigure is a linear fit equation. R² and *p* are two values to assess the relationship and significance between SCR and AAR.

We note that there are more SCR values than AAR values for Helm and Djankuat glaciers, which indicates that the SCR values can be used as a proxy for the missing AAR values. Abramov Glacier only has AAR data after 2012, but its SCR data extend back to 1985. Furthermore, Helm glacier yields the most similar trend between the SCR and AAR values and Silvretta, Gries, Vernagtferner, Kesselwandferner, Hintereisferner and Urumqi No. 1 glaciers also yield similar trends between the SCR and AAR values. However, the SCR values for Helm, Peyto, and Gries glaciers are overestimated.

### 4.2. Snow Line Altitude (SLA) Time Series

The annual maximum SLA and annual ELA values for the 14 reference glaciers are shown in Figure 8, along with their respective linear time-series trends. The maximum SLA is often lower than the ELA value, which is the opposite of the SCR–AAR relationship.

Helm, Place, Peyto, Silvretta, Gries, Djankuat, Abramov, Kesselwandferner, Hintereisferner, Urumqi No. 1 and Qiyi glaciers possess similar SLA and ELA trends, which means that the calculated SLA values represent the ELAs. However, there are inconsistencies in the SLA–ELA trends for Careser, Abramov, Vernagtferner, and Tsentralniy glaciers. The calculated SLA values for Careser Glacier do not match the ELA values and linear trend. There are only seven years of ELA data for Abramov Glacier, such that its SLA–ELA trend is sensitive to small differences between the SLA and ELA. The inconsistencies for Vernagtferner and Tsentralniy glaciers can be attributed to errors in the SLA calculations for some of the years that have skewed the linear SLA trend compared to the ELA trend. Kesselwandferner, Hintereisferner, Urumqi No. 1 and Qiyi glaciers possess good fits to the SLA and ELA data. For example, a very good match was obtained between the Qiyi SLA and ELA values in 1992, 1994, 1996, 1997, 1998, 2002, 2008, 2011, 2013, 2014 and 2015, with a calculated SLA–ELA difference of 24 m or less during these years. The largest difference for the entire time series was 225 m, which occurred in 2009, and the smallest difference was 4 m, which occurred in 2002.

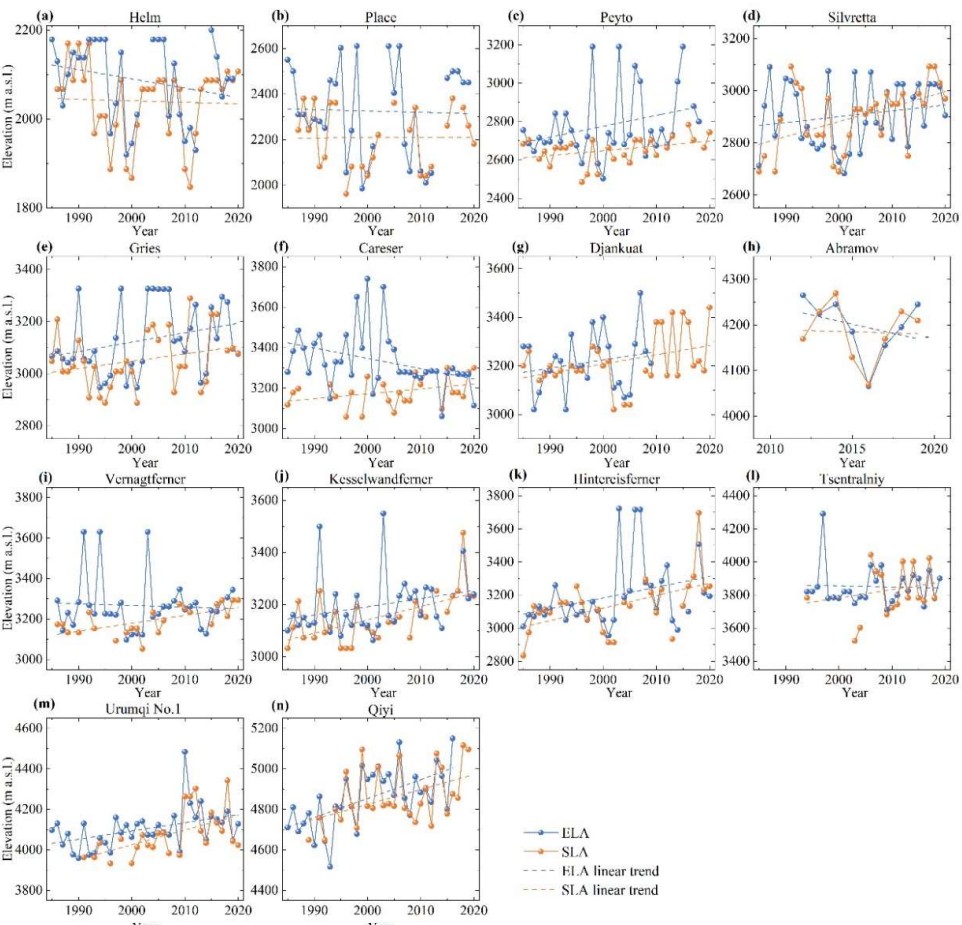

**Figure 8.** ELA and SLA time-series changes and linear trends for the 14 reference glaciers. Blue and orange dots represent the ELA and SLA values, respectively. Solid lines highlight the time-series changes, and dashed lines represent the linear trends. Subfigures (**a–n**) represent each value of SLA and ELA per year, and the time trend between the SLA and ELA of each glacier, respectively.

Figure 9 compares the maximum SLA and annual ELA values for the 14 reference glaciers. Hintereisferner Glacier possesses the highest $R^2$ value at 0.74 and significantly correlates at the 0.001 level, which means that the SLA provides an excellent fit for the ELA. Helm, Place, Silvretta, Djankuat, Kesselwandferner, Tsentralniy, Urumqi No. 1, and Qiyi glaciers possess $R^2$ values above 0.5 and significantly correlate at the 0.001 level. Gries

Glacier possesses the $R^2$ value at 0.44 and significantly correlates at 0.001 level. Besides, Peyto and Vernagtferner glaciers significantly correlate at 0.01 level and possess $R^2$ values at 0.32 and 0.40, respectively. Abramov glacier possesses $R^2$ values at 0.57 and significantly correlates at 0.05 level. The SLA derived from these three glaciers has a worse ability to indicate ELA than the nine glaciers mentioned above due to sparse SLA-ELA pairs, some huge SLA-ELA value differences, and other comprehensive reasons. However, these three glaciers also possess a good fit correlation between the SLA and ELA. Besides, Careser Glacier has not passed the significant test and only possesses an $R^2$ value of 0.003. The SLA values for Helm and Djankuat glaciers are larger than their corresponding ELA values, whereas Place, Peyto, Silvretta, Gries, Abramov, Kesselwandferner, Hintereisferner and Qiyi glaciers have similar SLA and ELA values. The calculated SLA values can serve as a proxy to estimate the ELA values for the years where the ELA is missing. The linear trends of Place, Gries, Djankuat, Kesselwandferner, Hintereisferner, Urumqi No. 1 and Qiyi glaciers yield a good-fitting SLA–ELA relationship.

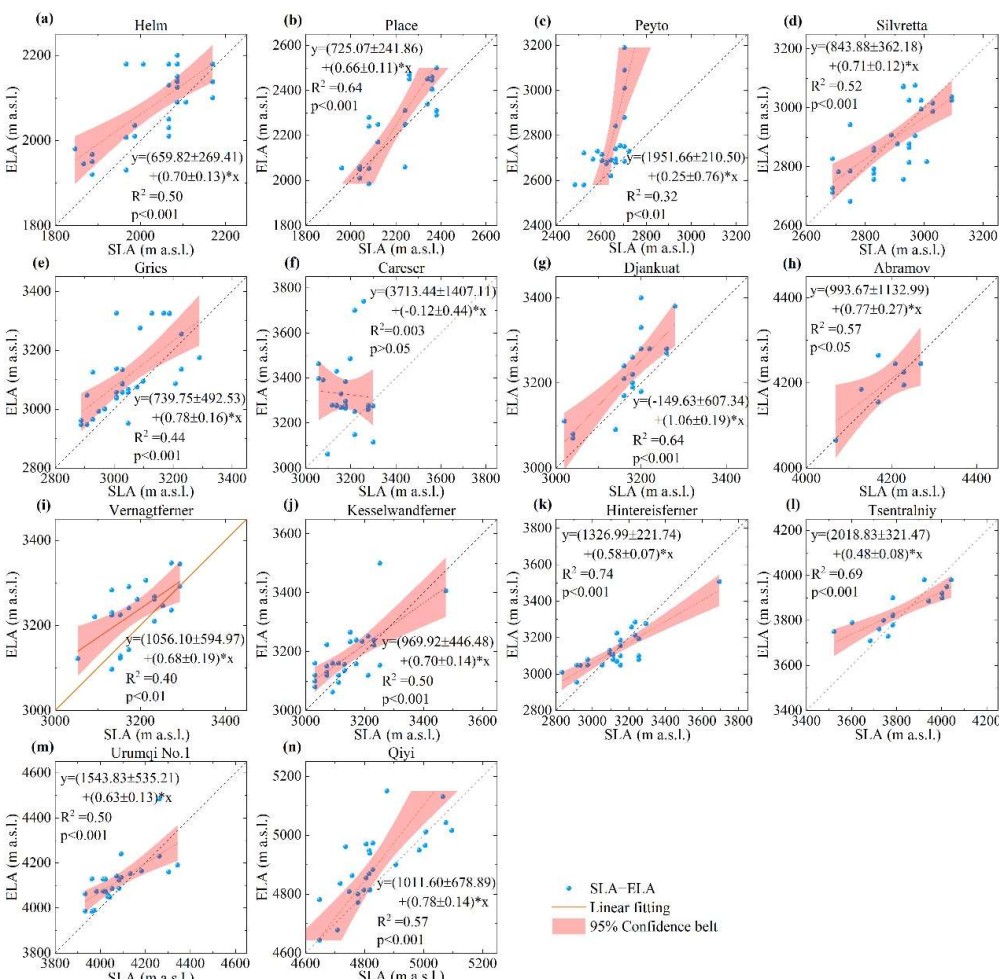

**Figure 9.** SLA–ELA relationships for the 14 reference glaciers. Blue points are the maximum SLA–annual ELA pairs, solid lines indicate the linear fits to the data, and red areas are the 95% confidence belts. Subfigures (**a–n**) represent the comparison of the relationship between the SLA and ELA of each glacier, respectively. The linear function equation in each subfigure is a linear fit equation. $R^2$ and *p* are two values to assess the relationship and significance between SLA and ELA.

## 5. Discussion

### 5.1. Snow Line Altitude (SLA)-Equilibrium Line Altitude (ELA) Comparison

The calculated annual SLA for each glacier is compared with its ELA to verify the algorithm. We know that each SLA result has a ±24 m uncertainty. We therefore calculate

the goodness of fit for each glacier. We define deviations of <24, 24–48, 48–96 and >96 m as very good, good, fit, and unfit SLA calculations. The results for each glacier are shown in Table 5 and the fitting situation in each region is shown in Figure 10. The Caucasus region has the highest very good fit rate (32%) and fit rate (37%), with 84% of the SLA data fitting with the ELA data. However, Djankuat Glacier is the only reference glacier in the Caucasus that was analyzed in this study, which means that the Djankuat Glacier data may be biasing the true state of the Caucasus region. The HMA and Alps SLA results also have good fits (without Careser Glacier), with the ELA data being fit by 74% of the SLA data. Western Canada has the poorest fit among the four regions, with an 18% very good fit rate, 17% good fit rate, and 31% unfit rate. Djankuat, Abramov, Kesselwandferner, Hintereisferner, Urumqi No. 1 and Qiyi glaciers all have comprehensive good-fitting rates. However, Peyto, Gries, and Careser have high unfitting rates.

**Table 5.** The matching situation of SLA and ELA per glacier.

| Glacier Name | Very Good Fit | Good Fit | Fit | Unfit | Glacier Name | Very Good Fit | Good Fit | Fit | Unfit |
|---|---|---|---|---|---|---|---|---|---|
| Helm | 5 | 7 | 12 | 6 | Abramov | 4 | 2 | 2 | 0 |
| Place | 5 | 3 | 7 | 8 | VNF * | 4 | 6 | 9 | 3 |
| Peyto | 4 | 3 | 8 | 10 | KWF * | 8 | 5 | 10 | 4 |
| Silvretta | 5 | 8 | 11 | 8 | HEF * | 7 | 5 | 6 | 8 |
| Gries | 4 | 9 | 6 | 12 | Djankuat | 6 | 3 | 7 | 3 |
| Careser | 4 | 2 | 3 | 16 | Qiyi | 7 | 5 | 5 | 9 |
| Tsentralniy | 1 | 5 | 6 | 4 | Urumqi No. 1 | 8 | 5 | 4 | 8 |

* VNF, KWF, HEF are Vernagtferner, Kesselwandferner and Hintereisferner, respectively.

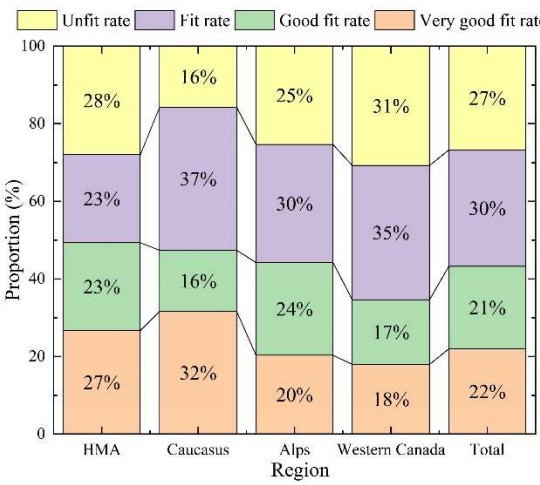

**Figure 10.** Glacier SLA–ELA matching situation for each region.

### 5.2. Algorithm Discussion

5.2.1. Glacier Outline Impact on the Snow Line Altitude (SLA) Calculation

The glacier outline is a key input parameter that can significantly impact the SLA calculation. An annual SLA calculation should be extracted from annual glacier outlines in the ideal situation, but such measurements are difficult to obtain from the published data. The NDSI and band ratios, with a threshold, can map the glacier extent in large areas, although this approach may be inaccurate for a single glacier. Therefore, the glacier outlines are derived from the RGI 6.0 database and published WGMS shapefiles. Some of the published glacier outlines are from the 1985, 2000, and 2015 periods, whereas other glaciers may have either more or less outline data. Here, we selected the latest glacier outline as an input parameter, which may make the SCR value lower since most glaciers

are experiencing accelerated melting and the most recent outline captures the smallest glacier extent. However, this method will ensure that rock and other land geometries do not participate in the snow and ice classification, and SLA calculation. However, errors will arise when a given glacier has recently undergone considerable changes. Careser Glacier has been experiencing accelerated mass loss for the past 40 years, with apparent variations in its length, area, and surface shape. Figure 11 shows its NIR band Landsat images in 1986, 1996, 2009 and 2020 and in its color Google Earth image. The western portion of Careser glacier has been almost melted, resulting in errors in the SLA calculation due to variations in the glacier outline. Rock, water and other landmarks will be classified as either snow cover or ice owing to the incorrect glacier outline. Here, we initially used the NDSI to calculate the glacier outline, but this approach failed because it could not delineate a precise glacier outline, leading to additional errors. In this study, we used the Careser Glacier outline from 2014 for the snow and ice classification, and SLA calculation, and we found that Careser Glacier possessed similar ELA and SLA values in 2011, 2014, 2015 and 2019, but possessed dissimilar ELA and SLA values before 2010. It highlights that glacier outline changes can have impacts on the SLA calculations. Besides, ELA measurements for Careser Glacier have reached over 3630 m a.s.l. while the highest altitude of Careser Glacier is 3278 m a.s.l., which also makes huge differences between ELA and SLA for Careser Glacier.

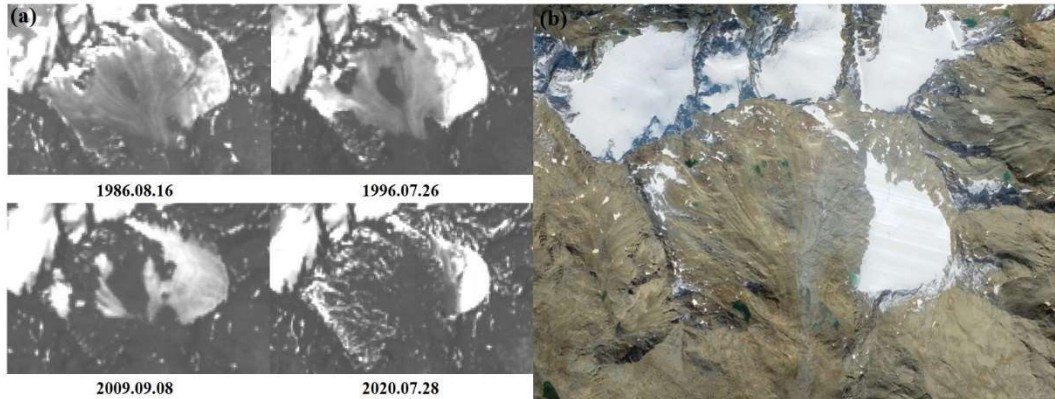

**Figure 11.** Careser Glacier changes during the 1986–2020 period. (**a**) The newNIR band Landsat images for Careser Glacier in 1986, 1996, 2009 and 2020. (**b**) Latest Google Earth image for Careser Glacier.

5.2.2. Algorithm Limitation

This study presents an automated algorithm to classify bare ice and snow cover on the glacier surface and then calculate the glacier SLA in GEE. However, two key processing limitations exist: (a) cloud cover and shadow impacts and (b) the Otsu classification method limitation.

Cloud cover and shadow are direct challenges for snow and ice classification of the glacier surface using optical images. The natural land surface cannot be detected when cloud covers an optical image, which means some useful data are missing. Affected pixels lead to misclassification and affect SCR estimations and further SLA calculations (example shown in the Supplementary Materials).

Here, we selected only those images in which >65% of the useful area along the total glacier extent was preserved. Therefore, cloud cover and shadow may still occur but may not seriously affect some of the images; however, it will cause some wrong classification results because the Otsu method will classify cloud cover as ice owing to its low reflectance in the new NIR band. We tried to remove the effective area by cloud and used the remaining area for our classification, but we found that worse results were obtained because the total area had changed. This indicated that the SCR was invalid, and that some void areas would interfere with the algorithm to select the correct zone for the SLA calculation. Therefore,

we used cloud cover and shadow detection instead of cloud cover removal because the cloud cover impacts were small in the used images (shown in Figure 12).

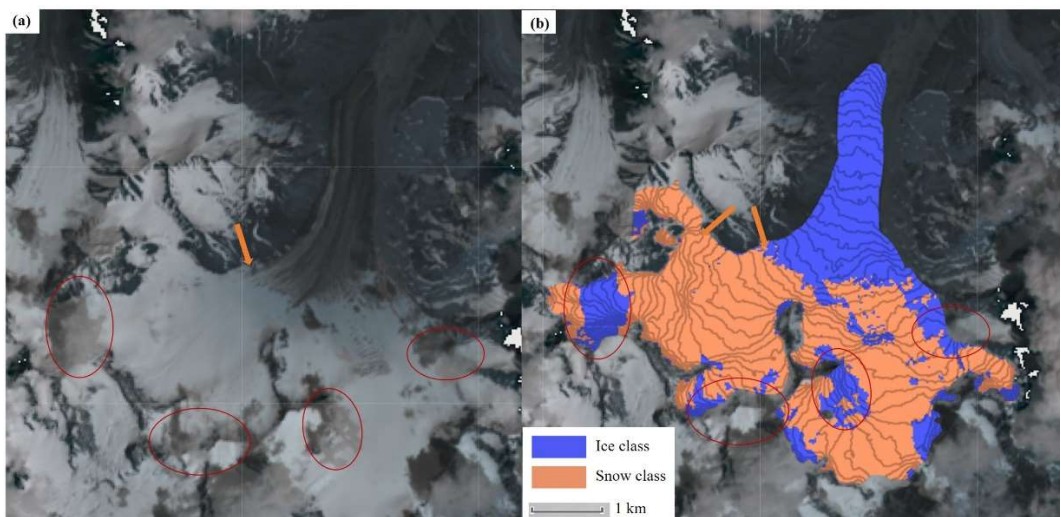

**Figure 12.** Abramov Glacier snow and ice classification, and SLA calculation. (**a**) Landsat image for Abramov Glacier in the NIR band, with some cloud cover and shadow affects. (**b**) Snow and ice classification for Abramov Glacier. Some misclassifications are marked by red circles; however, the SLA is unaffected. Orange arrows in (**a**,**b**) indicate the correct SLA calculations.

The Otsu method can be applied correctly when there is an apparent difference between the bare ice and snow cover across the analyzed glaciers. However, an incorrect classification result may be obtained when snow covers the entire glacier or there is limited snow cover on the glacier. In the first situation, only one type of surface geometry will be obtained when fresh snow covers the entire glacier extent: snow. Nevertheless, the Otsu method will classify the dark areas, such as the edges of the image and rock as ice. Here, we use the new NIR band to improve the adaptation to this situation and obtain a better result than using the single NIR band. Besides, an image where the entire glacier is dominated by fresh snow cover does not represent the SLA for that year. Conversely, an image with minimal snow cover means intense ablation for that season. Some areas will be incorrectly classified as snow cover, resulting in an underestimation of the SLA. We tested other classification methods, such as the random forest method, but there was no improvement in the results because different images possess different bright light conditions, which means the sample data for one image may not fit in another image.

### 5.2.3. Landsat Series Images Impact on the Snow Line Altitude (SLA) Calculations

Here, we analyze images from four main satellite sensors (TM sensor in Landsat-4 and 5, ETM+ sensor in Landsat-7, and OLI sensor in Landsat-8) during the 1985–2020 period for the SLA calculations, which introduces two issues: (a) different Landsat image matching questions; and (b) the impact of combining different Landsat images on the SLA results.

We used the Level 2, Collection 2, Tier 1 Landsat series products, which are registered, atmospherically corrected, orthorectified SR products with a 50-m global geolocation accuracy. The alignment of the images was visually confirmed by loading four sensor images at one point, we found that there was no need for additional registration. Besides, we unified the type during the preprocessing step and selected the same data type to ensure that different images would not affect the results. However, the Landsat-7 images possessed a scan line corrector error after 2003, with black and void stripes covering the regions in the images that possessed no data. This error may affect the image selection because no data area may reduce the glacier area.

The other issue is that the combination of various Landsat images can yield different temporal resolutions. The Landsat revisit time is 16 days; however, cloud cover or other factors that affect the image can increase the revisit time to 32 days or even longer. Accelerated glacier melting can increase the local atmospheric humidity and enhance the probability of cloud cover during strong ablation years. Therefore, the actual maximum SLA day may be missing if successive Landsat images are not obtained. This is one reason why some high ELA years do not possess a good match with the calculated SLA. Conversely, a combination of various Landsat images may yield better SLA results than a single Landsat image calculation. For example, a joint analysis of Landsat-5 and Landsat-7 images may yield a shorter revisit time, as more images are available to calculate the SLA. We therefore calculated the average ELA–SLA deviation during two periods in Table 6 (1985–1999 and 2000–2020) for each analyzed glacier. The year 2000 was selected as the time that separated the two periods since the Landsat-7 satellite was launched in 1999; we consider that the post-2000 SLA calculations were made using data from two sensors. Helm, Place, Peyto, Silvretta, Vernagtferner, Kesselwandferner, Tsentralniy, and Urumqi No. 1 glaciers all possess a lower average ELA–SLA deviation during the 2000–2020 period compared with the 1985–1999 period. Furthermore, the average ELA–ELA deviations for the 1985–1999 and 2000–2020 periods are 50.18 and 39.68 m, respectively (Table 6). We therefore believe that multiple satellite sensors will generate more images and facilitate more precise glacier SLA calculations than a single satellite sensor.

**Table 6.** Average ELA–SLA deviations during two time periods for thirteen of the reference glaciers.

| Glacier Name | Average ELA-SLA Deviation (1985–1999)/m | Average ELA-SLA Deviation (2000–2020)/m | Glacier Name | Average ELA-SLA Deviation (1985–1999)/m | Average ELA-SLA Deviation (2000–2020)/m |
|---|---|---|---|---|---|
| Helm | 59.07 | 44.50 | VNF * | 55.89 | 35.16 |
| Place | 53.09 | 36.14 | KWF * | 64.15 | 18.43 |
| Peyto | 125.69 | 110.42 | HEF * | −8.58 | 11.50 |
| Silvretta | 17.17 | −15.40 | Tsentralniy | 37.00 | 21.06 |
| Gries | 61.53 | 67.41 | Urumqi No. 1 | 59.60 | 39.25 |
| Djankuat | 48.46 | 58.33 | Qiyi | 29.10 | 75.56 |
| Abramov | NA | 13.50 | Total Average | 50.18 | 39.68 |

* VNF, KWF, HEF are Vernagtferner, Kesselwandferner and Hintereisferner, respectively.

### 5.3. Further Works

In this article, fourteen reference glaciers have been calculated and verified the correlation between SCR-AAR and SLA-ELA. In the entire algorithm process, the cloud cover, shadow and the Otsu method limitation are analyzed how these factors together influence the SLA calculation. However, some challenges and problems still need to be discussed and solved in further works.

(a) Debris-covered glaciers' SLA calculation should be researched. For global mountain glaciers, debris-covered glaciers over 7.3% of Earth's mountain glacier area, presenting a significant factor contributing to the variability of glacier response to climate changes in different regions [42–45]. Debris-covered area and thickness affect the glacier ablation process, affecting the calculation of AAR and ELA. Besides, the composition of debris will also affect the snow ablation process, leading to impacts on SCR and SLA. Therefore, the relationship between SCR-AAR and SLA-ELA needs further research on debris-covered glaciers.

(b) Quantifying the impact of the limitations on SCR and SLA should be researched. This article analyzed the total effect of mentioned limitations and factors between the SCR-AAR and SLA-ELA relationship. It is still difficult to quantify how each factor affects the SCR and SLA calculation individually.

(c) Multiple satellite datasets should be incorporated into the algorithm to reduce images' temporal resolution, leading to the true days of SLA closed to the day of ELA. We are trying to add some Sentinel-2 images in Landsat series images to do further testing.

## 6. Conclusions

This study proposed an automated algorithm for classifying the bare ice and snow cover on a given glacier and calculating the annual glacier SLA during the 1985–2020 period in GEE. Fourteen reference glaciers in HMA, the Caucasus, Alps, and Western Canada were investigated to test and validate the algorithm. The calculated SCR and SLA values were verified by the measured AAR and ELA values, respectively.

The algorithm performed well for all of the reference glaciers, with the exception of Careser Glacier, with a glacier SLA uncertainty of approximately ±24 m. Ten glaciers possessed a high correlation ($R^2 > 0.5$) between the calculated SLA and measured ELA values, and eight of these glaciers also possessed a high correlation ($R^2 > 0.5$) between the calculated SCR and measured AAR values, which indicated a good fit between these data. Three of the remaining glaciers possessed decent correlations ($R^2 > 0.3$) between the SLA and ELA values, and five of the remaining glaciers possessed decent correlations ($R^2 > 0.3$) between the SCR and AAR values, which highlighted a good relationship between the field observations and remote-sensing-derived calculations.

The glacier outline is a significant component in the SLA calculation process. The glacier outline for some glaciers that are undergoing accelerated melt, such as Careser Glacier, has experienced considerable surface changes, such that the calculated SLA yields a poor fit to the true ELA. We used the latest published glacier outline as an input parameter for the other glaciers and obtained good results.

From a regional perspective, the Caucasus glacier yielded the best fit between the SLA and ELA, with the HMA and Alps glaciers also yielding good fits. The Western Canada glaciers yielded worse results than the other three regions. From a time perspective, the average ELA–SLA deviation for the 1985–1999 period was higher than that for the 2000–2020 period. Merging multiple Landsat datasets increased the number of optical images and shortened the temporal resolution, which allowed the calculated SLA to be closer to the true ELA.

However, the algorithm still possesses key limitations. Cloud cover and shadow block useful information, thereby yielding incorrect snow and ice classifications in some instances. Furthermore, the Otsu method is prone to misclassification errors when either fresh snow covers the entire glacier or there is minimal snow cover across the glacier. Future work will employ the algorithm to calculate the glacier SLA in HMA to obtain SCR and SLA for nearly 100,000 glaciers. We will also incorporate multiple satellite datasets, including Sentinel-2 and SAR data, to improve the accuracy of the results.

**Supplementary Materials:** The following supporting information can be downloaded at: https://www.mdpi.com/article/10.3390/rs14102377/s1.

**Author Contributions:** Software, X.L.; writing—original draft preparation, X.L.; supervision, N.W.; validation, Y.W.; funding acquisition, N.W. All authors have read and agreed to the published version of the manuscript.

**Funding:** This article was supported by the Strategic Priority Research Program of the Chinese Academy of Sciences (Grant No. XDA19070302 and XDA20060201), the National Natural Science Foundation of China (Grant No. 42130516), the Second Tibetan Plateau Scientific Expedition and Research Program (Grant No. 2019QZKK020102), and the National Natural Science Foundation of China (Grant No. 42171139).

**Data Availability Statement:** The data presented in this study are available on request from the corresponding author.

**Acknowledgments:** We thank the USGS and Google for the use of available satellite imagery. We would also like to thank WGMS and NCDC for supplying glaciers' data.

**Conflicts of Interest:** The authors declare no conflict of interest.

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
