# Peer review of "Automated Glacier Snow Line Altitude Calculation Method Using Landsat Series Images in the Google Earth Engine Platform"

_remotesensing, doi:10.3390/rs14102377_

Round 1

Reviewer 1 Report

The authors have addressed my comments and I am satisfied with the improvements.

Reviewer 2 Report

The Authors  modified the manuscript according to the suggestions of reviewers. This new version of the manuscript is more straightforward and the adopted procedures clearer. As suggested, the authors have also added further references to support the methodology adopted for image processing. The result is a more comprehensive and complete paper.

 I suggest to publish the paper in its current version

This manuscript is a resubmission of an earlier submission. The following is a list of the peer review reports and author responses from that submission.

Round 1

Reviewer 1 Report

The paper presents an interesting approach for an automated analysis of big data retrieved by satellite optical sensors available on the GEE infrastructure. The presentation sounds simplified since SLA and SCR are strictly connected in the data chain and limitations are discussed even if they are treated in the description of the specific steps. How do those limitations impact on SCR estimations and consequently on the SLA assessment? Results are overestimated using not-statistically defined metrics: specific tests can be explored to state if R2 of 0.5 is a good fit.  

Figure 1 – font too small and a line connecting points to the specific zoom box could be helpful. The orange glacier outline doesn’t help, please change the colour…

Figure 2 – Is the SLA estimation derived by the SCR calculation? Lines 312-313 suggest a link between SCR and SLA, not two independent chains. The SLA before and after the 2020 diamond sounds like something different? Is it the same for SCR? Maybe it is better to include a Time series analysis in the middle between the year check and the annual estimations?

Lines 196-199 How do the selected thresholds has been defined and what is the impact of such selections (65% of the glacier area and 3 images per year) on the final analysis?

Lines 219-221 What do you mean with “manually revised”?

Section 3.3 What is the overall accuracy of the SCR estimation? This critical step lack in such an assessment: what is the efficiency on discriminating snow and ice?

Figure 4 – font too small please remove RGB colour components from the legend: it is better to declare the use of false colour background using the combination R (which SWIR band from Table 3) + G (Green from Table 3) + B (Blue from Table 3). Which Platform has been considered? Which date?

Lines 311, 373, 418 – I would see acronym definition in titles Snow line altitude (SLA) instead of SLA. Please check the journal author guidelines.

Section 4.1 The AAR is used as validation dataset for the SCR time series, can authors give more details about AAR, information in the introduction gives few details about the methods and the interpretation. It is important for the reader to know why they are using AAR as validation dataset. What about statistical tests assessing the significance of the relationships between AAR and SCR? I read “excellent” and “good-fitting”, it is an opinion or a statistical assessment?

Section 5.1 How do the fit classes have been defined? The fit class seems to be selected with a SLA below 96 meters (4 times the uncertainty) from the ELA estimation. Is it possible to define statistically such a threshold?  Why 96 meters is a good result? Maybe is it an acceptable fit?

Reviewer 2 Report

The authors present an automated algorithm to classify bare ice and snow cover on glaciers using Landsat series images, calculate the minimum annual glacier snow cover ratio (SCR) and maximum SLA for world-wide reference glaciers during the 1985–2020 period in Google Earth Engine. The use of a big data algorithm for this purpose is the main novelty of the article. The objectives, data and methodology are well presented, and the cartographic material illustrate the main results. Also, the article is well structured. However, the uncertainty of the method is quite important and the statistical relationship between snow cover parameters is modest (for 3 glaciers is only 30 % to 50 % of total variation is explained (r2 > 30 %).

This study has a very god potential to be published after the discussion of some issues described below:

  1. The authors make no reference to the part of the glacier covered with debris. Studies show that especially in the Caucasus (https://doi.org/10.1017/jog.2021.47 ) these areas are important. How the authors consider that their presence influences the quality of the results of this study.
  2. The cause of the modest relationship between the parameters of the snow layer must be discussed in more detail
  3. The authors specify whether the script for automated algorithm written in JavaScript in GEE will be available to the scientific community after the article is published.

Reviewer 3 Report

In this work, the Landsat image collection of fourteen glaciers is processed to extract the glacier snow line altitude after discriminating bare ice from snow cover in the images.

The authors seem to have taken only partial account of the literature on the recognition of snow from ice in Landsat multispectral images. The idea of using an automatic technique for image processing is correct but Landsat images should be pre-processed with more accuracy.

The illumination conditions of satellite images must be taken into account since, considering the morphology of the glaciers examined; shadows from the reliefs can also be an issue for snow/ice detection. The bibliographical references reported by the authors also highlight this aspect, but it seems that the authors do not consider it; in any case, there is any indications in the text whether and how the effects linked to the shadows of the reliefs are considered in the processing procedures.

In my opinion, the evaluation of error is also approached in an approximate manner; the concept of mixed pixels is introduced but the problem is only superficially resolved.

Authors follow the method proposed in ref.[38and 39] and apply it to different glaciers.It is not clear, however, how the glaciers were selected and whether their geomorphological and / or climatic characteristics (e.g. extension, altitudes, lithology of the surrounding soils etc.) were taken into account. This does not make the considerations in the results paragraph clear.

The correlations between the identified ratios (AAR.SCR, SLA, ELA elevation etc.) should be better discussed taking into account also the environmental conditions of the examined glaciers in order to support the considerations presented in the conclusions.

Furthermore, the description of the algorithm in section 5.2.2. does not help to understand the processing procedures and does not exhaustively clarify the approach adopted to overcome the limitations of the algorithm described.

Considering positively the basic idea behind this work, once the required major revisions have been made, the work can be accepted for publication.

 More specific remarks:

Row 72: Why according to the authors NDVI is not useful for classification ? it needs to be better argued.

Row 89-90: This paragraph is not clear. Clarify and possibly add references for the method described.

Row 114-115: Add a reference for “Otsu method”; it will help to understand the adopted classification criteria

Row 123: Why the authors selected these glaciers? Only because of the available time series? If yes, some environmental (climate) information should also be included in order to evaluate the effectiveness of the analyses. 

Row196: Is the shadow of the mountains included in “other factor”?  Even if the images are orthorectificated, the reliefs in these area are not negligeable and some shadow could be present due to the illumination conditions at the moment of images acquisition.

Row295: The reflectance of debris is related to their lithology and water in NIR and SWIR bands do not have “bright condition”. If this happens in what the author call “NIR new band”, please add an example.

Row 320: I think that a reference on the “zone method” is necessary; only the “point method” is described in ref [23].

Row 323: How can a DEM with 20 m interval be adapted to an image of 30 m per pixel? Please add some further information and references.

Row 339: How can Landsat images (pixels=30m) produce a 1:25000 scale output? Please explain and add references

Row 362: Mixed pixels can be processed with different methodological approaches (unmixed methods). Please support your methodological choice.  

Row 367: How does the 15m uncertainty relate to what you report in line 360 where the uncertainty is considered to be +- 1 pixel (30m)?

Row 474 - Tab 4: This analysis should include considerations about morphological and environmental conditions of the glacier (for example, did the extent of the glacier affect the goodness of the fit?)

Row 493: Why is the NIR band used in fig. 10 and fig.12 and not the NIR new? The NIR new is indicated as the one actually used for the classification so it should be the most suitable to highlight the differences.